

# A preliminary study of the salivary microbiota of young male subjects before, during, and after acute high-altitude exposure

Qian Zhou[1,2,*], Yuhui Chen[2,*], Guozhu Liu[3], Pengyan Qiao[2] and Chuhua Tang[1,2]

[1] The fifth Clinical Medical College of Anhui Medical University, Clinical College of Anhui Medical University, Beijing, China
[2] Department of Stomatology, PLA Strategic Support Force Medical Center, Beijing, China
[3] The 32183 Military Hospital of PLA, Baicheng, Jilin, China
[*] These authors contributed equally to this work.

## ABSTRACT

**Background**. The microbial community structure in saliva differs at different altitudes. However, the impact of acute high-altitude exposure on the oral microbiota is unclear. This study explored the impact of acute high-altitude exposure on the salivary microbiome to establish a foundation for the future prevention of oral diseases. Methods. Unstimulated whole saliva samples were collected from 12 male subjects at the following three time points: one day before entering high altitude (an altitude of 350 m, pre-altitude group), seven days after arrival at high altitude (an altitude of 4,500 m, altitude group) and seven days after returning to low altitude (an altitude of 350 m, post-altitude group). Thus, a total of 36 saliva samples were obtained. 16S rRNA V3-V4 region amplicon sequencing was used to analyze the diversity and structure of the salivary microbial communities, and a network analysis was employed to investigate the relationships among salivary microorganisms. The function of these microorganisms was predicted with a Phylogenetic Investigation of Communities by Reconstruction of Unobserved States (PICRUSt) analysis.

**Results**. In total, there were 756 operational taxonomic units (OTUs) identified, with 541, 613, and 615 OTUs identified in the pre-altitude, altitude, and post-altitude groups, respectively. Acute high-altitude exposure decreased the diversity of the salivary microbiome. Prior to acute high-altitude exposure, the microbiome mainly consisted of Proteobacteria, Firmicutes, Bacteroidetes, Fusobacteria, and Actinobacteria. After altitude exposure, the relative abundance of *Streptococcus* and *Veillonella* increased, and the relative abundance of *Prevotella*, *Porphyromonas*, and *Alloprevotella* decreased. The relationship among the salivary microorganisms was also affected by acute high-altitude exposure. The relative abundance of carbohydrate metabolism gene functions was upregulated, while the relative abundance of coenzyme and vitamin metabolism gene functions was downregulated.

**Conclusion**. Rapid high-altitude exposure decreased the biodiversity of the salivary microbiome, changing the community structure, symbiotic relationships among species, and abundance of functional genes. This suggests that the stress of acute high-altitude exposure influenced the stability of the salivary microbiome.

Corresponding author
Chuhua Tang, tch306@126.com

## INTRODUCTION

The number of people travelling to high altitudes has increased in recent years due to the availability of transportation as well as increased travel for work, sports events, and even earthquake relief (*Seccombe & Peters, 2014*; *Hodkinson, 2011*). In some cases, travellers may experience acute high-altitude exposure, which occurs when individuals ascend to areas over 2,500 m above sea level within 24 h. High-altitude environments have low atmospheric pressure, low oxygen levels, large temperature differences, and strong ultraviolet radiation (*Lackermair et al., 2019*; *Torlasco et al., 2020*; *Swenson et al., 2020*). Therefore, acute exposure to high-altitude environments can lead to negative health consequences, such as increased blood pressure and increased platelet aggregation, and may even result in acute high-altitude sickness (*Revera et al., 2017*; *Lackermair et al., 2019*; *Luks, Swenson & Bartsch, 2017*). Evidence suggests that acute high-altitude exposure may also induce oral diseases. For example, patients with no previous history of toothache developed symptoms of acute pulpitis within 48 h of flying to high-altitude locations; this onset is thought to be related to high-altitude hypoxia, decreased atmospheric pressure, and external stimulation of dental pulp after dental caries (*Zhang, 2020*). Periodontitis, gingivitis, and oral ulcers are the most common oral diseases reported upon travel to high altitudes (*Li & Liu, 2018*). Additionally, in one study, rats developed more serious periodontal damage under simulated high-altitude hypoxic conditions (*Xiao et al., 2012*). Moreover, while oral diseases can directly result in oral symptoms, they can also indirectly increase the risk of systemic diseases such as myocardial infarction, infective endocarditis, and respiratory diseases (*Sampaio-Maia et al., 2016*).

The aetiology of oral diseases induced by high-altitude environments is complex. Microorganisms are the initiating factor in oral infections and thus are closely related to the occurrence of disease (*Arweiler Nicole & Netuschil, 2016*). In healthy individuals, the oral microbial community maintains a dynamic balance, which provides protection against adverse external stimulation. An imbalance in microorganisms or between microorganisms and the host increases the risk of dental caries and periodontitis (*Zhang et al., 2018*). The microorganisms in oral plaque biofilm are considered to be the pathogenic bacteria in the development of periodontal disease. These pathogenic bacteria include the red complex bacteria (*Tannerella forsythia*, *Treponema denticola*, and *Porphyromonas gingivalis*), which drive the pathogenesis of periodontal disease by regulating the recombination of microbiota and promoting the inflammatory response (*Valm, 2019*). In addition, previous studies have found that oral microbiota is associated with the development of oral cancer and primary tumors beyond the head and neck (*Mascitti et al., 2019*; *Zhang et al., 2018*; *Stasiewicz & Karpiński, 2022*; *Liu et al., 2022*). Epidemiological studies have shown that periodontal disease is associated with an increased risk of esophageal, gastric, pancreatic, and colorectal cancers (*Michaud et al., 2017*; *Hu et al., 2018*; *Fan et al., 2018*). This may be related to dysbiosis of the oral microbiota leading to the development of chronic inflammation,

altered metabolic activity leading to the increased production of toxic metabolites and immune responses that promote tumorigenesis and tumor growth (*Snider, Freedberg & Abrams, 2016*; *Inaba et al., 2014*; *Gainza-Cirauqui et al., 2013*).

Altitude, atmospheric oxygen levels, temperature, psychological stress, disturbances in circadian rhythms, and sleep deprivation all affect the composition, distribution, and metabolic activity of oral microorganisms (*Gao et al., 2018*; *Lamont, Koo & Hajishengallis, 2018*; *Liu et al., 2021*). The oral cavity has multiple unique niches and a symbiotic bacterial ecosystem. The five major niches of the oral cavity are the saliva, tongue surface, oral mucosa, tooth surface, and subgingival plaque. Each niche has a different microbial community (*Consortium, 2012*). Saliva contains more than 700 different microbial species, which makes saliva a potential pool of biomarkers (*Wang et al., 2022*). In orthodontic patients, the diversity of the salivary microbiome was lower at high altitudes; the relative abundance of *Streptococcus* in the salivary microbiome increased, while the relative abundance of *Veillonella* decreased (*AlShahrani et al., 2020*). These studies suggest that the salivary microbial community differs between high- and low-altitudes, but no studies have investigated whether acute high-altitude exposure influences the composition and functional metabolism of the salivary microbial community.

Therefore, the aim of this study was to examine changes in the composition and structure of the salivary microbial community after acute high-altitude exposure and to explore the correlation between bacteria and bacterial functions. The results of this study expand the current understanding of the influence of high-altitude exposure on salivary microorganisms, providing a theoretical foundation for the future development of preventive measures for oral diseases associated with high altitudes.

## MATERIALS & METHODS

### Study subjects

The study protocol was approved by the ethics review board of the PLA Strategic Support Force Medical Center (No. K2021-10). Written informed consent was obtained from all study participants. All of the procedures were performed in accordance with the Declaration of Helsinki and with all relevant policies in China.

The subjects were recruited from a scientific research unit in a city in northern China. The inclusion criteria for the subjects were as follows: (1) ethnic Han male, over 18 years old; (2) long-term resident of the low-altitude plain; (3) no oral infectious diseases; (4) no systemic or genetic diseases; and (5) no history of smoking. The exclusion criteria were as follows: (1) subjects who had travelled to high-altitude locations within the past six months; (2) subjects who could not tolerate a high-altitude environment; (3) subjects who had received antibiotics, hormones, or other drugs within the past 3 months; and (4) subjects who had received periodontal treatment within the past 3 months. A total of 12 healthy male subjects aged from 26 to 45 years (mean age 34.9 ± 7.0 years) were recruited for this study, and all subjects agreed to participate.
## Sample collection

This study examined the salivary microbiome of 12 subjects who were transferred from a plain (at an altitude of 350 m) to a plateau (at an altitude of 4,500 m) within a period of 12 h. All subjects stayed in a hotel at high altitude for seven days. The intensity and duration of their physical activity (walking) were similar, but these data were not captured. Unstimulated whole saliva samples were collected from each subject at three different time points: one day before reaching the plateau (pre-altitude group), seven days after entering the plateau (altitude group) and seven days after returning to the plain (post-altitude group). A total of 36 saliva samples were obtained.

All participants were required to fill in a unified oral health questionnaire before and after entering the plateau. In addition, in order to minimize the impact of other interference factors on saliva samples, the following requirements were made for the subjects during the study: (1) the diet of each subject was uniformly provided by the study organization; (2) before the start of the study, a professional stomatologist taught all study subjects about the pasteurization tooth brushing method, and tried to ensure that brushing method and time were the same for all study participants; (3) a uniform toothbrush and toothpaste was used by all study participants. Samples were collected from 9 a.m. to 11 a.m., which is consistent with previous studies (*Gill, Price & Costa, 2016*). During sample collection, the subjects first rinsed their mouth with deionized water and then were instructed to sit quietly for 10 min. Then, a disposable sterile saliva collection tube was placed on the oral mucosa of the subject's lower lip, allowing saliva to naturally flow into the collector. After approximately 2 ml of saliva had been collected, a preservation solution was added to the collection tube, and the sample was stored in a −20 °C refrigerator, and then transferred to a lab at low altitude through cold chain transport (liquid nitrogen) to be stored at −20 °C for further study (*Gill, Price & Costa, 2016*).

## Bacterial DNA extraction and PCR

Total genomic DNA from the samples was extracted using the cetyltrimethylammonium bromide (CTAB) method (*Kachiprath et al., 2018*). DNA concentration and purity were assessed on 1% agarose gels (Beijing, China). Then, the DNA concentration was diluted to 1 ng/µL with sterile water. Specific barcode primers were used to amplify the DNA in the 16S V3-V4 sequencing region with high-efficiency, high-fidelity enzymes (Phusion® High-Fidelity PCR Master Mix, New England Biolabs, Ipswitch, MA, USA). All PCR products were run on 2% agarose gels for detection, and the PCR products were purified with a Qiagen Gel Extraction Kit (Qiagen, Hilden, Germany).

## DNA library preparation and sequencing

Sequencing libraries were generated using the TruSeq® DNA PCR-Free Sample Preparation Kit (Illumina, San Diego, CA, USA) following the manufacturer's recommendations; index codes were also added (*Yang et al., 2019*). FLASH (V1.2.7 http://ccb.jhu.edu/software/FLASH/) was used to splice reads from each sample to obtain raw tags, which were rigorously filtered to obtain high-quality tags (*Magoc & Salzberg, 2011*) using the QIIME (V1.9.1 http://qiime.org/scripts/split_libraries_fastq.html) operation

procedure (the default quality threshold was <= 19; *Bokulich et al., 2013*). The resulting library was sequenced on an Illumina NovaSeq platform, and 250 bp paired-end reads were generated.

## Data analysis

All effective tags were clustered into operational taxonomic units (OTUs) by the UPARSE software (V7.0.1001 http://drive5.com/uparse/, 97% identity; *Sevinsky et al., 2010*). Taxonomic information and community composition were obtained through OTU annotation analysis (*Haas et al., 2011*; *Edgar, 2013*). A Venn diagram was used to define the shared and unique microbes at the species level by OTU clustering. Alpha and beta diversity were analysed to investigate differences in microbial community structure among groups (*Simons et al., 2019*). Additionally, a principal coordinate analysis (PCoA) was performed to obtain principal coordinates and visualize complex, multidimensional data (*Wang et al., 2020*). Analysis of similarity (ANOSIM) was used to determine differences in community structure among groups and to compare the differences within and between groups (*Yang et al., 2019*). A network analysis was used to determine the relationships among dominant genera by calculating the correlation coefficient (*Niquil et al., 2020*). Based on the abundance of bacteria, a PICRUSt analysis was used for the functional prediction of microbial communities in the saliva samples (*Douglas, Beiko & Langille, 2018*).

## Statistical analysis

SPSS 24.0 software (SPSS Inc., Chicago, IL, USA) was used for the data analyses. The Shapiro–Wilk test was performed sequentially to ascertain the normality of the distribution of the data. If $P$ values <0.05, the alpha diversity, beta diversity, taxa, and metabolism gene functions were evaluated using non-parametric (Wilcoxon) tests. If $P$ values >0.05, the alpha diversity, beta diversity, taxa, and metabolism gene functions were evaluated using Student's $t$-tests. $P$ values <0.05 were considered statistically significant.

# RESULTS

## Basic information of study subjects

After entering the plateau, three subjects showed symptoms of gingival swelling and pain, and two patients showed symptoms of oral ulcers. These symptoms resolved upon returning to low altitude and none of the 12 subjects reported negative oral symptoms seven days after returning from the high altitude (Table S1).

## Global sequencing data

In total, 3,138,051 raw sequences were generated from the 36 saliva samples, with an average of 87,168 raw sequences per sample (Table S2). Quality filtering was used to acquire 2,298,989 effective sequences, with an average of 63,861 sequences per sample. The shortest sequence length of the effective tags was 418 bp, the longest sequence length was 425 bp, and the average sequence length was 422 bp (Table 1).

Clustering of all effective sequences was based on a threshold of 97% identity. These sequences were then annotated against the Silva138 database to determine species

**Table 1** The assembly results from the saliva samples.

| Sample number | Raw PE | Raw tags | Clean tags | Effective tags | Effective (%) | Min length (bp) | Max length (bp) | Average Length (bp) |
|---|---|---|---|---|---|---|---|---|
| 36 | 3,138,051 | 2,944,826 | 2,802,821 | 2,298,989 | 73.64% | 418 | 425 | 422 |

**Notes.**

(1) Sample number refers to the 36 samples; (2) Raw PE is the first data read out by the sequencing platform; (3) Raw tags refers to the sequence after splicing primer fragments from the original data; (4) Clean tags indicates a high-quality sequence with an appropriate length obtained by filtering raw tags; (5) Effective tags refers to the sequences in clean tags that do not have chimeras and can be used for in-depth analysis; (6) Effective % refers to the ratio of effective data to the original number of offline users %; (7) Min length refers to the minimum length of valid data; (8) Max length refers to the maximum length of valid data; (9) Average length refers to the average length of valid data.

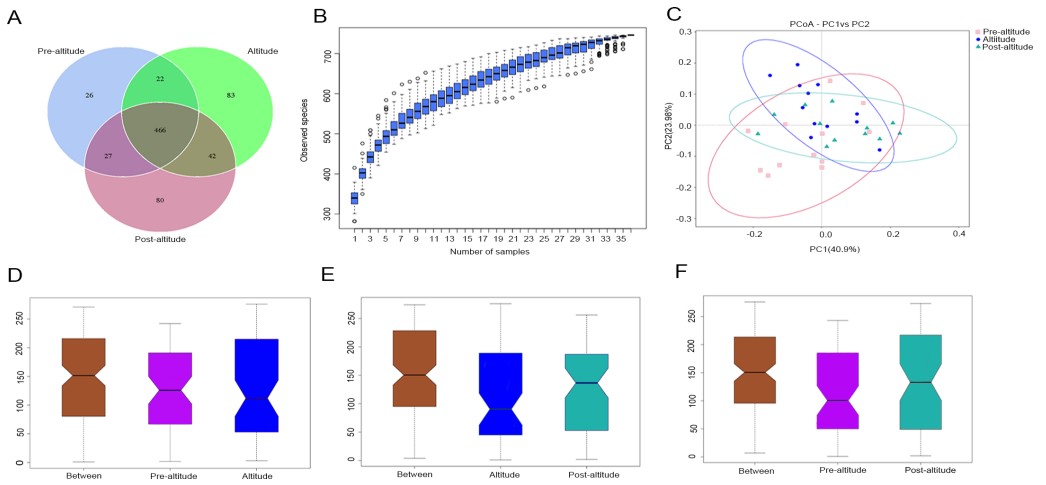

**Figure 1** **Comparisons of microbial community structure among the three groups.** (A) Venn diagram of OTUs shared between the different groups. (B) Species accumulation box diagram of saliva samples. (C) Weighted UniFrac-PCoA of salivary microbiota from the three groups. PC1 explained 40.90% of the variation observed, and PC2 explained 23.98% of the variation. (D–F) Analysis of similarity (ANOSIM) of community structure differences among the three groups. The ordinate shows the rank of the distance between the samples. The abscissa labelled "Between" shows the result between two groups, and the other two are the results within their respective groups. The comparison between any two groups found that $R > 0$, $P < 0.05$, indicating that when the difference between groups is greater than that within the individual groups, the difference is statistically significant (Figure1D: $P = 0.03$, $R = 0.1379$. Figure1E: $P = 0.002$, $R = 0.2127$. Figure1F: $P = 0.004$, $R = 0.2452$).

taxonomy. In total, there were 756 OTUs identified, of which 653 could be annotated with the database. The number of annotated OTUs at the phylum level was 625, and the number of annotated OTUs at the genus level was 520.

In the Venn diagram, we identified 541, 613, and 615 OTUs in the pre-altitude, altitude, and post-altitude groups, respectively. These groups shared a total of 466 OTUs. A total of 26, 83, and 80 unique OTUs were found in the pre-altitude, altitude, and post-altitude groups, respectively (Fig. 1A).

## Bacterial diversity analysis

The species accumulation curves tended towards saturation, indicating that the amount of sample was sufficient (Fig. 1B). Based on the normality test, the alpha diversity was

**Table 2  Bacterial alpha diversity indices for saliva samples in each group.**

| Group | Chao1 | | ACE | | Shannon | | Inverse Simpson | | Simpson even | | Good's coverage | |
|---|---|---|---|---|---|---|---|---|---|---|---|---|
| | Mean | SE | Mean | SE | Mean | SE | Mean | SE | Mean | SE | Mean | SE |
| Pre-altitude | 361.64 | 4.07 | 362.62 | 4.70 | 5.75[b] | 0.08 | 26.40[d] | 1.74 | 0.08[e] | 0 | 0.99 | 0 |
| Altitude | 359.44 | 11.03 | 363.40 | 11.59 | 5.42[a,b] | 0.11 | 19.31[c,d] | 1.56 | 0.06[e] | 0 | 0.99 | 0 |
| Post-altitude | 375.43 | 6.95 | 379.19 | 7.35 | 5.74[a] | 0.07 | 25.28[c] | 1.72 | 0.07 | 0 | 0.99 | 0 |

Notes.
[a]Shannon index between Altitude and Post-altitude group indicated a statistically significant difference ($P = 0.024$).
[b]Shannon index between Pre-altitude and Altitude group indicated a statistically significant difference ($P = 0.025$).
[c]Inverse Simpson index between Altitude and Post-altitude group indicated a statistically significant difference ($P = 0.022$).
[d]Inverse Simpson index between Pre-altitude and Altitude group indicated a statistically significant difference ($P = 0.007$).
[e]Simpson even index between Pre-altitude and Altitude group indicated a statistically significant difference ($P = 0.004$).

evaluated using Student's $t$-tests and the beta diversity was evaluated using non-parametric (Wilcoxon) tests between groups. The ACE and Chao1 richness indexes were higher in the post-altitude group than in the pre-altitude and altitude groups, but these differences did not reach significance. The Shannon and inverse Simpson diversity indexes were significantly lower in the altitude group than in the pre-altitude and post-altitude groups. Simpson's evenness index was higher in the pre-altitude group than in the altitude group, indicating that the bacterial community distribution in the salivary samples was very uneven. In addition, Good's coverage index was 99.9% for each group, indicating that the sequencing depth was sufficient to detect the bacterial diversity of the saliva samples (Table 2).

Based on the weighted UniFrac distances, the PCoA of beta diversity showed that the samples formed well-separated clusters corresponding to the three groups, suggesting that the oral microbiota community structure differed among the three groups (Fig. 1C). The ANOSIM based on the Bray–Curtis distances of the salivary microbiota structure revealed significant differences between the pre-altitude and altitude groups (Fig. 1D), the altitude and post-altitude groups (Fig. 1E), and the pre-altitude and post-altitude groups (Fig. 1F). Thus, differences in oral microbiota community structure between each group were observed.

## Bacterial abundance and distribution

The predominant bacteria were largely consistent among the three groups, but differences in relative abundances were observed. The ten predominant phyla included Proteobacteria, Firmicutes, Bacteroidetes, Fusobacteria, Actinobacteria, Spirochaetes, Tenericutes, unidentified Bacteria, Cyanobacteria, and Synergistetes (Fig. 2A). Based on the normality test, the relative abundance of taxa was evaluated using Student's $t$-tests between groups. Among the predominant phyla with a mean relative abundance >1% in each group, the relative abundance of Firmicutes was significantly higher and that of Bacteroidetes was significantly lower in the altitude group than in the pre-altitude group, and the relative abundance of Actinobacteria was significantly higher and that of Bacteroidetes was significantly lower in the post-altitude group than in the pre-altitude group. Compared to the altitude group, the relative abundance of Firmicutes was significantly lower in the post-altitude group (Figs. 2B–2D).

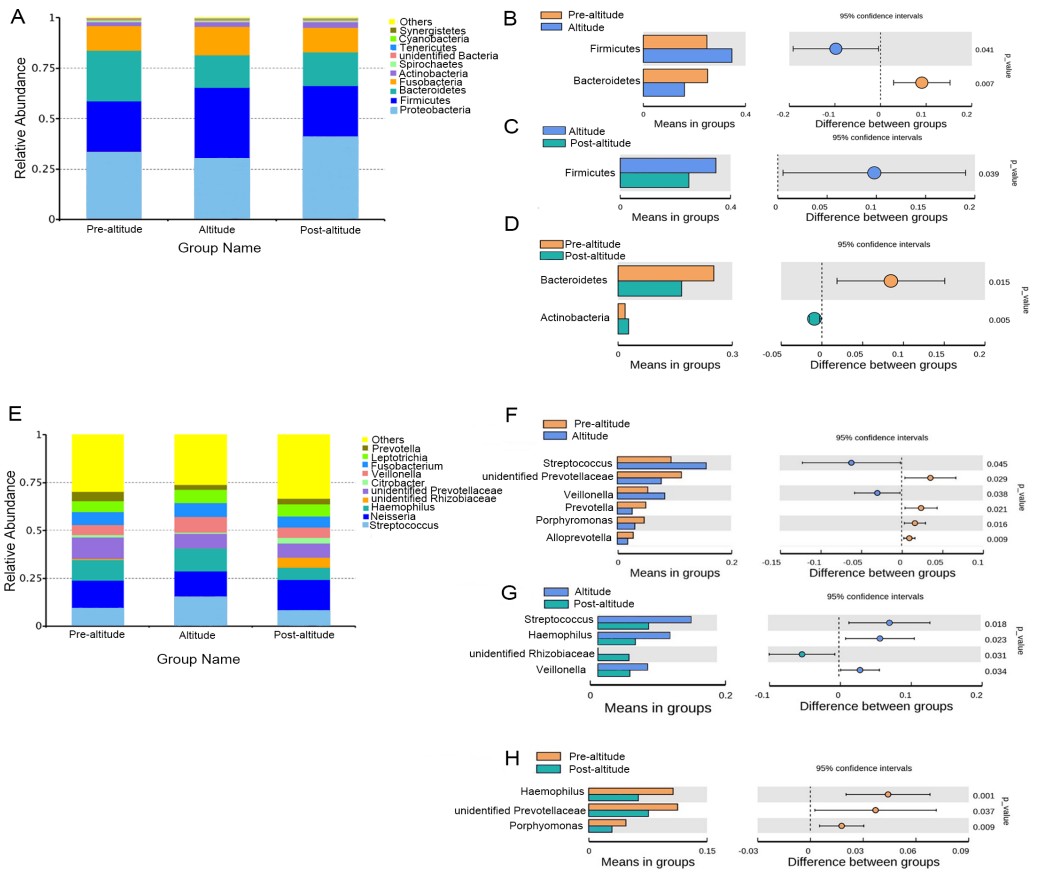

**Figure 2  Bacterial abundance and distribution.** (A) Distribution of the ten predominant bacteria at the phylum level. (B–D) The predominant phyla showing more than 1% of the mean relative abundance in the pre-altitude, altitude, and post-altitude groups. (E) Distribution of the ten predominant bacteria at the genus level. (F–H) The predominant genus showing more than 1% of the mean relative abundance in the pre-altitude, altitude, and post-altitude groups. Statistically significant differences are marked with *P*-values.

The ten predominant genera included *Streptococcus*, *Neisseria*, *Haemophilus*, *unidentified Prevotellaceae*, *unidentified Rhizobiaceae*, *Citrobacter*, *Veillonella*, *Fusobacterium*, *Leptotrichia*, and *Prevotella* (Fig. 2E). Among the predominant genera with a mean relative abundance >1% in each group, a significantly higher relative abundance of *Streptococcus* and *Veillonella* were observed in the altitude group compared to the pre-altitude group, whereas a significantly lower relative abundance of *unidentified Prevotellaceae*, *Prevotella*, *Porphyromonas* and *Alloprevotella* were observed in the altitude group compared to the pre-altitude group. Additionally, compared to the pre-altitude group, a significantly lower relative abundance of *Haemophilus*, *unidentified Prevotellaceae*, and *Porphyromonas* were observed in the post-altitude group. Compared to the altitude group, the relative abundance of *Streptococcus*, *Haemophilus* and *Veillonella* were significantly lower in the post-altitude group, while the relative abundance of *unidentified Rhizobiaceae* was significantly higher

in the post-altitude group (Figs. 2F–2H). These genera changed in more than half of the subjects.

In addition, we found that three subjects developed gingival swelling and pain symptoms and two subjects developed oral ulcer symptoms after entering the plateau. However, no significant difference was found between the symptomatic and asymptomatic subjects in the changes of oral dominant bacteria (Fig. S1).

### Network analysis

A network analysis was used to explore the bacterial co-occurrence patterns and provide insight into interactions among the salivary microbiota. There were 736 edges and 91 nodes in the pre-altitude group and 601 edges and 89 nodes in the altitude group. The pre-altitude group had an average path length (APL) of 3.363, a network diameter (ND) of 9 and a graph density (GD) of 0.074. The structural properties of the pre-altitude group network differed from those of the altitude group network (Table S3).

The network diagram for the 100 predominant bacteria at the genus level were shown in pre-altitude and altitude groups (Figs. S2, S3). There was a higher relative abundance of *Streptococcus* in the pre-altitude group and altitude group. *Streptococcus* was strongly correlated with *Veillonella*, *Prevotella* and *Fusobacterium* in the pre-altitude group, and with *Neisseria* and *Haemophilus* in the altitude group.

### PICRUSt function predictions

A PICRUSt analysis was performed to predict the potential functions of the salivary microbiota. The ten predominant gene functions were predicted in Kyoto Encyclopedia of Genes and Genomes (KEGG) level 1, and the most predominant gene function was found to be metabolism (Fig. 3A). The ten predominant gene functions were then predicted in KEGG level 2 (Fig. 3B). The pre-altitude and altitude groups had different KEGG profiles in hierarchy level 3. Based on the normality test, the relative abundance of metabolism gene functions was evaluated using Student's t-tests between groups. Ten gene functions were significantly different between the pre-altitude group and the altitude group. Notably, the genes involved in carbohydrate metabolism were upregulated in the altitude group, and the genes involved in the metabolism of cofactors and vitamins were downregulated in the altitude group (Fig. 3C). These results indicate that the gene function of the salivary microbiota changed upon acute high-altitude exposure.

## DISCUSSION

Rapidly ascending to a high altitude has been shown to have negative effects on health and may also contribute to the development of oral diseases. However, while the salivary microbiome of residents at high altitudes is known to differ from that of residents at low altitudes, the effects of acute high-altitude exposure on the salivary microbiome remain unclear. This study showed that acute high-altitude exposure decreased the diversity of the salivary microbiome and influenced the relationships among salivary microorganisms. The function of the microbiome was also altered by high-altitude exposure: genes involved in carbohydrate metabolism were upregulated, while genes involved in coenzyme and vitamin

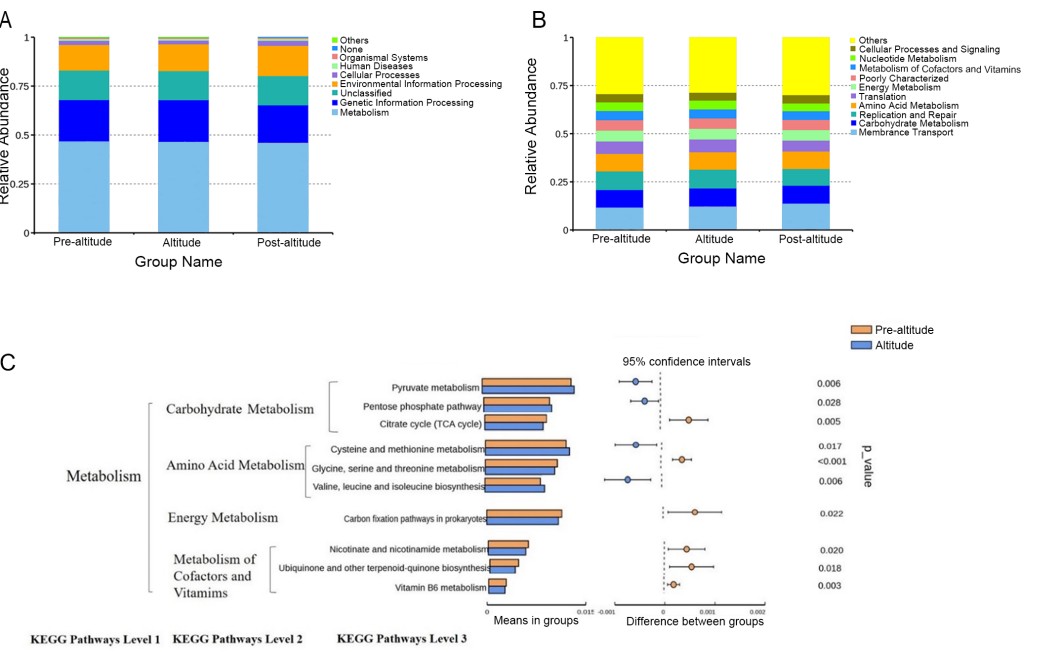

**Figure 3** **PICRUSt function predictions.** (A–B) Distributions of the top ten predicted gene functions of the salivary microbiota in the pre-altitude, altitude, and post-altitude groups are shown at KEGG Pathways Level 1 (A) and at KEGG Pathways Level 2 (B). (C) Statistically significant gene functions in a comparison of the predicted gene functions of salivary microbiota between the pre-altitude and altitude groups at KEGG Pathways level 3.

metabolism were downregulated. These results provide insight into the development of oral diseases at high altitudes.

The relationship between the salivary microbiome and oral and systemic health has been widely studied (*Duran-Pinedo, 2021*; *Sampaio-Maia et al., 2016*). Previous studies have found that participants experienced symptoms of oral discomfort, such as spontaneous gingival bleeding and tooth pain, within seven days of travelling to a high-altitude location (*Li & Liu, 2018*). Therefore, we selected three time points for saliva collection: one day before travelling to a high altitude, seven days after arriving at a high altitude, and seven days after returning to a low altitude. This allowed us to analyze and compare the oral microbiomes at these three time points. Saliva sample collection is simple, non-invasive, and low-cost.

The diversity of the oral microbial community can reflect oral health. A previous study reported that the alpha diversity of the salivary microbial community in patients with dental caries was lower than that in healthy people (*Belstrom et al., 2017*). In our study, the alpha diversity of the salivary microbial community was lower in the altitude group than in the pre-altitude and post-altitude groups, which suggests that the prevalence of oral diseases may increase after high-altitude exposure. In addition, *Liu et al. (2021)* reported that, compared with a Tibetan population living in high-altitude areas, the alpha diversity of the salivary microbial community in a population living at ultrahigh altitudes decreased and that there was a significant negative correlation between altitude and the alpha diversity of

the salivary community. *AlShahrani et al. (2020)* found a lower diversity of oral microbes in orthodontic patients at high altitudes than in orthodontic patients at low altitudes; patients at high altitudes were also more susceptible to periodontitis. Moreover, exposure to hypoxia has been shown to reduce the secretory function of salivary glands, resulting in the destruction of periodontal tissue (*Terrizzi et al., 2018*). We speculate that the alpha diversity of the salivary microbiome changes after high-altitude exposure because the colonization of oral bacteria is affected by oxygen concentration.

In this study, anaerobic bacteria were dominant in the saliva of the altitude group and inhibited the colonization of aerobic bacteria. This may have led to the observed decrease in the alpha diversity of the salivary microbiome. Our results also showed that the beta diversity of the salivary microbial community changed after high-altitude exposure. Previous studies have found a positive correlation between the beta diversity of the salivary microbial community and altitude, indicating that beta diversity may also be affected by altitude (*Bhushan et al., 2019*). Overall, these results indicate that acute high-altitude exposure affects the alpha and beta diversity of the salivary microbial community.

Changes in salivary symbiotic bacteria and increases in pathogenic bacteria may lead to the development of oral diseases (*AlShahrani et al., 2020*). A study of Tibetan residents living at different altitudes found that the relative abundance of Firmicutes decreased and that of Bacteroides increased with increased altitude (*Liu et al., 2021*). *Bhushan et al. (2019)* found that the relative abundance of Firmicutes decreased and that of Bacteroides increased 25 days after subjects entered the Antarctic. However, in this study, Firmicutes increased significantly and Bacteroides decreased significantly after moving to a high-altitude plateau. Our study also found that the relative abundance of *Streptococcus* and *Veillonella* increased after acute high-altitude exposure. These results are similar to those of another study that found that the relative abundance of *Streptococcus* appeared to increase after acute high-altitude exposure (*Zhao et al., 2022*). These two genera are thought to be involved in the formation of oral plaque, which is implicated in the occurrence and development of dental caries and periodontal disease (*Marsh & Zaura, 2017*). However, (*Liu et al., 2021*) reported a decreased relative abundance of *Streptococcus* with altitude, and *AlShahrani et al. (2020)* found higher levels of *Streptococcus* and lower levels *of Veillonella* in orthodontic patients at high altitudes than in those at low altitudes. In this study, the relative abundance of *Prevotella*, *Porphyromonas*, and *Alloprevotella* decreased after acute high-altitude exposure. A previous study also showed that the relative abundance of *Prevotella* was higher at high altitudes (*Xiao et al., 2012*). There are no reports on the effect of altitude on *Alloprevotella* and *Porphyromonas* in the oral cavity. However, one study reported that the relative abundance of *Alloprevotella* in the stomach of cattle increased with altitude (*Fan et al., 2020*). The above studies all suggest that high-altitude environments affect the abundance of common dominant bacteria, but some of the results are inconsistent with those of this study. These differences may be related to differences among studies in the time spent at high altitude.

Although this study did not find significant differences in the microbiota between subjects with swelling and aching of the gums (or oral ulcer) and asymptomatic subjects, it clearly demonstrated the bacterial community and its symbiosis pattern in the saliva of

the subjects under acute high-altitude exposure. As this was an observational study, it is not clear whether the change in bacteria was directly affected by altitude differences or the interactions among microorganisms. Therefore, we believe that acute high-altitude exposure affects oral microbial homeostasis to a certain extent. This finding is helpful to understanding the potential impact of acute high-altitude exposure on human oral health, but the specific mechanism needs to be further studied.

Oral symbiotic microbes are important for maintaining oral health. Symbiotes promote oral health through resistance to colonization by pathogens; the symbiotes outperform disease-causing species in the colonization matrix and thus have little chance of integration by exogenous pathogens (*Chalmers et al., 2008*; *Thurnheer & Belibasakis, 2018*). It has been reported that *Streptococcus* and *Actinomyces* isolated from the oral environment of healthy people can inhibit the growth of *Porphyromonas gingivalis* (*Sedghi et al., 2021*). In our study, the correlation between *Streptococcus* and *Haemophilus*, *Veillonella*, and *Prevotella* changed after rapid high-altitude exposure. Therefore, this study found changes in the symbiotic relationship of oral microbes, suggesting that altitude may break down the protective barriers these symbiotic microbes create.

Microorganisms not only participate in the body's immune response but also affect metabolic activity (*Martin et al., 2010*). Our data showed that the gene functional gene profile of the oral microorganisms was significantly affected by acute high-altitude exposure. Genes involved in carbohydrate metabolism were predicted to be upregulated, which may indicate that the oral microflora had increased energetic demands in the high-altitude environment. In addition, genes involved in the metabolism of cofactors and vitamins, including vitamin B6, were predicted to be downregulated in the altitude group. Previous studies have shown that a high level of vitamin B6 can help the human body eliminate reactive oxygen species, prevent oxidative stress damage, and adapt to a harsh external environment (*Hellmann & Mooney, 2010*). Moreover, other studies have shown that the vitamin B6 pathway is upregulated in the skin microbiota of high-altitude populations (*Li et al., 2019*) and in the oral microflora of an ultrahigh-altitude population compared with that of a high-altitude population (*Liu et al., 2021*). However, *Monnoyer et al. (2021)* found that in the extreme environment of saturation diving, the abundance of aerobic metabolic pathways in the oral bacteria of divers increased, while the anaerobic metabolic pathways—mainly energy metabolism, oxidative stress, and adenosine cobalamin synthesis—decreased. Our results also found that the expression of genes related to vitamin metabolism was downregulated after participants were taken to high altitudes. Based on these findings, we speculate that acute high-altitude exposure may affect the relative abundance of gene functions related to oral bacterial metabolic activity and may aggravate oxidative stress damage. These changes may affect oral health. As our results are based only on the predicted functions of the salivary microbiota, they do not represent the actual functions of the oral bacteria. Further analysis of the roles of these genes in the oral cavities of people exposed to high altitudes is needed.

This study investigated the effects of acute high-altitude exposure, an environmental stressor, on the salivary microbiome. Since our study was only preliminary, we plan to use larger sample sizes, clinical indicators, and more accurate monitoring of subjects'

physical activity in future studies. Future work should explore the relationship between oral microbiota and oral health.

## CONCLUSIONS

In conclusion, the diversity of the salivary microbiota decreased after acute high-altitude exposure, with an increase in the relative abundance of *Streptococcus* and *Veillonella* and a decrease in the relative abundance of *Prevotella*, *Porphyromonas*, and *Alloprevotella*. In addition, the correlation between bacteria genera changed. Bacterial metabolic functions include an increase in the relative abundance of carbohydrate metabolism gene functions and a decrease in the relative abundance of coenzyme and vitamin metabolism gene functions. These results advance our understanding of the salivary microbiota at high altitudes and its influence on oral diseases. Future research should explore the specific mechanism of the effect of acute high-altitude exposure on the homeostasis of oral microorganisms.

## ACKNOWLEDGEMENTS

We acknowledge Beijing Nuohe Zhiyuan Technology Co., Ltd for their kind help performing the Illumina NovaSeq PE250 sequencing and data analysis using the NovaSeq6000 Platform.

### Funding

This work was supported by a grant from Military Medical Science and Technology Youth Cultivation Program (17QNP033). The funders had no role in study design, data collection and analysis, decision to publish, or preparation of the manuscript.

### Grant Disclosures

The following grant information was disclosed by the authors:
Military Medical Science and Technology Youth Cultivation Program:  17QNP033.

### Competing Interests

The authors declare there are no competing interests.

### Author Contributions

- Qian Zhou conceived and designed the experiments, performed the experiments, analyzed the data, prepared figures and/or tables, authored or reviewed drafts of the article, and approved the final draft.
- Yuhui Chen conceived and designed the experiments, performed the experiments, authored or reviewed drafts of the article, and approved the final draft.
- Guozhu Liu performed the experiments, authored or reviewed drafts of the article, and approved the final draft.

- Pengyan Qiao performed the experiments, authored or reviewed drafts of the article, and approved the final draft.
- Chuhua Tang conceived and designed the experiments, performed the experiments, authored or reviewed drafts of the article, and approved the final draft.

## Human Ethics

The following information was supplied relating to ethical approvals (i.e., approving body and any reference numbers):

The PLA Strategic Support Force Characteristic Medical Center granted Ethical approval to carry out the study within its facilities (No. K2021-10).

## Field Study Permissions

The following information was supplied relating to field study approvals (i.e., approving body and any reference numbers):

The PLA Strategic Support Force Characteristic Medical Center granted Ethical approval to carry out the study within its facilities (No. K2021-10).

## Data Availability

The sequences are available at NCBI: PRJNA841273.

Available at https://www.ncbi.nlm.nih.gov/bioproject/PRJNA841273.

## Supplemental Information

Supplemental information for this article can be found online at http://dx.doi.org/10.7717/peerj.15537#supplemental-information.

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
