# Peer review of "A preliminary study of the salivary microbiota of young male subjects before, during, and after acute high-altitude exposure"

_PeerJ, doi:10.7717/peerj.15537_

## Round 0.1 · original submission · Major Revisions

Dear Dr. Zhou and colleagues:

Thank you for submitting your manuscript to PeerJ. I have now received three independent reviews of your work, and as you will see, the reviewers raised some concerns about the research. I encourage you to revise your manuscript accordingly, considering all of the concerns raised by the three reviewers.

Reviewer 1 ·

Basic reporting

Manuscript title: “A preliminary study of the salivary microbiota in young male subjects under acute high-altitude exposure”.
This is an interesting study investigating the changes in the salivary flora during acute altitude exposure.
I have some considerations:
1. Only minor language corrections should be necessary to improve readability.
2. I suggest adding some considerations about the relationship between an altered oral microbiota and the risk of developing tumors, suggesting an even more important role of periodontal pathogens in human health (doi: 10.3389/fcimb.2019.00232).

Experimental design

By using 16S rRNA sequencing technique, the Authors analysed the effects of acute high-altitude exposure on the oral microflora. Although the sample studied was small, this study represents an interesting starting point for future development of preventive measures for acute high-altitude oral diseases.

Validity of the findings

The results were interesting, showing that acute altitude exposure decreased the diversity of the salivary microbiome (e.g., increase of Streptococcus and Veillonella species, decrease of Prevotella, Porphyromonas, and Alloprevotella species). Furthermore, the expression of genes involved in carbohydrate metabolism was upregulated, while the expression of genes involved in coenzyme and vitamin metabolism was downregulated.

Additional comments

no comment

Reviewer 2 ·

Basic reporting

1. Overall, more context is needed throughout the introduction/discussion. The authors begin to draw a link between the oral microbiota, high-altitude environments, and various oral diseases however more effort is needed in the introduction drawing specific links between changes in the microbiota and oral diseases. If less is known about these specific links, then the authors are encouraged to unpack their rationale for the study. Why or what would specific changes in the microbiota lead to the development of oral diseases (i.e., what is the mechanism)? Could decreases in microbiota diversity lead to greater incidence of upper respiratory tract infections? Or perhaps contribute to the development of oral ulcers?

2. The authors should re-review the literature cited throughout the introduction. In some instances, the authors inappropriately cite articles to support their statements. For example, on line 69 the authors cite Karl et al. 2018 to support their statement regarding the effect of various perturbations on the oral microbiota. However, Karl et al. 2018 explored the gut microbiota. If the authors are attempting to draw parallels between the gut and oral microbiota this should be explicitly stated and rationalized.

3. Some of the discussion simply states the results rather than attempting to explain or place the findings in the larger body of literature. For example, lines 334 to 343 restates the results but does not explore the meaning of those results. What is the significance between correlations in these bacteria?

4. Figure and tables are effective. I was unable to view the raw data with the URL supplied by the authors.

Experimental design

1. The research is primary (original) and well within the scope of the journal. In addition, the authors clearly state their primary purpose, though their hypotheses are not clearly stated (the research is exploratory in nature).

2. More information is needed regarding the statistical analyses. When did the authors chose to use parametric (t tests) vs non-parametric (Wilcoxon) tests?

3. Sample size should be justified, if possible, how did the authors arrive on a sample of 12? In addition, could the authors explain why the study was limited to only male participants?

Validity of the findings

1. In the discussion the authors state that their findings provide insight into the development of oral disease at high altitude. Contrary to this statement, the authors report no relationship between microbiota changes and reported symptoms. If the data do provide such insight the authors are encouraged to draw specific links to their findings and potential clinical impact.

2. The authors should limit their conclusions by linking them to specific supporting results. For example, in the conclusion the authors state that “microbial composition, diversity, community structure, networks, and gene functions of the salivary microbiota were affected by acute high-altitude exposure.” Specifically, how were these factors affected?

Additional comments

Zhou et al. investigated the impact of acute high-altitude exposure on salivary microbiota composition in a cohort of young males. Findings reveled that rapid exposure to 4500 m altered salivary microbiota, however these changes did not seem to be related to some of the adverse symptoms (gingival swelling and pain or formation of oral ulcers). The study is an interesting preliminary investigation that explores the interaction between altitude and microbiota, though the small sample size and lack of diversity (all males subjects) limit the extrapolation of the findings.

·

Basic reporting

In this study, the authors investigated the impact of acute exposure to high altitude on the salivary microbiome to establish a foundation for the prevention of oral diseases. After collecting saliva samples from 12 male subjects, they identified changes in the relative abundance of Firmicutes and Bacteroidetes after altitude exposure (4500 meters). At the genus level, they observed an increase in the relative abundance of Streptococcus and Veillonella, and a decrease of Prevotella, Porphyromonas, and Alloprevotella. The results are interesting as they indicate a change in bacterial behavior that authors link to the hypoxic conditions. Further analyses were done on the oral microbiota functions and the existing relationships among the different bacterial genera.

In general, the literature gives sufficient background in the introduction on the notion of high altitude and interest in studying the oral microbiota related to oral health. The discussion is well-written and the references are coherent with the research question. Some references are in my opinion not necessarily well-suited and Figures are not always appropriately described and labeled.
More comments on this are made in section 4 'Additional comments'.

Experimental design

In order to investigate the effect of acute high altitude exposure on the oral microbial structure: saliva samples were collected from 12 male subjects before, during, and after a stay at 4500 meters of altitude.

I had some trouble understanding the setup of the sample collection. It is a bit unclear how many days the subjects spent at the plateau. Did they spend 7 days at the plateau (with the sample collection just before returning to the plain)? Were the subjects in a facility? What was their physical activity during their stay at high altitude? Explanations could be added in the Sample collection section of Materials and Methods.
Were the samples collected at high altitudes stored at -20°C or transferred at -20°C after arrival at the plain?

Validity of the findings

The authors found that at acute high altitude exposure, the alpha diversity of the oral microbiota of the subjects decreased with the domination of anaerobic bacteria in the altitude group (Firmicutes and Bacteroidetes). Did the authors observe no changes in the relative abundance of aerobic bacteria (Proteobacteria and Actinobacteria) between pre and altitude groups?
Beta diversity was assessed by principal coordinate analysis (PCoA) based on the weighted UniFrac distances and an analysis of similarity (ANOSIM) based on the Bray–Curtis distance.
The predictive functional analysis of the oral microbial communities using PICRUSt showed a change in some metabolic pathways including carbohydrate, coenzyme, and vitamin metabolism. Let’s not forget that this analysis is based on inferences from 16S rRNA sequences. Therefore, I would suggest the authors include the notion of prediction in the abstract: either in the methods or in the results.
Regarding the link between high altitude and oxidative stress, did the authors measure the respiratory rate in the subjects? Do you think this could have led to a dry mouth which would have had a negative impact on the oral microflora and therefore the oral bacterial communities within it?
Finally, the network analysis showed that the relationship among the salivary microorganisms was affected by acute altitude exposure. What was the number of edges and nodes for the pre and altitude groups?
The results are well explained in the discussion. However, I think the diversity figures lack clarity in their understanding, and comments on the figures are listed in section 4.

Besides, I had a question about the validity of the ANOSIM results. P-values for the ANOSIM analysis are given in the legend of Figure 1 but not the one for R, which is needed, only that 'R>0'. Otherwise, as stated by Anderson et al, 2013: 'If a statistically significant result is accompanied by a value of R that is not very large, this could be a signal that the difference is primarily a difference in dispersion'.

-Conclusions are well stated, and linked to the original research question:
Many previous studies have explored the link between high altitude/extreme environments and oral microbiota structure and metabolic functions. This study is interesting because the authors compare their results with the ones observed on subjects that are living or working in extreme environments (i.e Tibetans or Antarctic), or having orthodontic appliances (orthodontic patients). Therefore, they are direct observations of how the oral microbiota is affected by a sudden change in oxygen concentration. In this regard, it would have been nice to emphasize the notion of the oxygen requirements of the bacteria. Maybe a comparison with observations made on oral microbiota in other extreme environments such as hyperbaria hyperoxia like in saturation diving could also be added to the discussion.

Additional comments

Abstract
Lines 20-21: Maybe the authors could precise the meters of altitude for the pre and altitude groups.

Introduction
Line 69: The authors say 'oral microorganisms' and refer to a paper on gut microbiota.
Line 75: maybe another reference would be more appropriate.
Lines 76-78: I think there is confusion when describing the results from Alshahrani et al. that could lead to wrong interpretations from the authors in their discussion.
In the original paper, Alshahrani et al. wrote that 'At the genus level, Streptococcus constituted more than 85 %, Veillonella around 10 % and Desulfobacta less than 5 % in Group 1 (high altitude). In Group 2 (sea level), Streptococcus (30 %), Veillonella (30 %), Bacillus (15 %), and Prevotella (10 %) were the four predominant genera seen'.
However, the authors wrote 'In orthodontic patients, the diversity of the salivary microbiome was lower at high altitudes; the relative abundance of Streptococcus in the salivary microbiome decreased, while the relative abundance of Veillonella and Prevotella increased (Alshahrani, 2020)'. I agree with the first part of the sentence, however, the rest are observations in the orthodontic patients at sea level compared to those at high altitude.
Besides, in the discussion (lines 313-314), the authors wrote ' Alshahrani et al. found lower levels of Streptococcus and Veillonella in orthodontic patients at high altitudes than in those at low altitudes'.
I thought Streptococcus was higher in orthodontic patients at high altitudes?

Results
Line 207-209: I think Tenericutes is missing from the list of the ten predominant phyla (based on the list of Figure 2.A).

Discussion / References
Line 269: in the bracket with the three references, there is a '2012' alone. Is it a typo?
Lines 307-309: Is (Mishiro et al., 2018) the right reference here? I don’t see that this study was performed at a high altitude.
Line 279: maybe another reference on adults would be more appropriate.
Line 316: A reference is written 'Xiao et al., 2012b'. It means that there is another publication with this author's name that was published in the same year because the authors added 'b'. However, I can’t find this second reference in the list. Besides, the reference (Xia, 2012) in line 58 should then be changed accordingly with the one in the list of references written as 'Xiao et al., 2012a'.
The same goes for the reference 'Gill et al., 2016' where the authors sometimes wrote '2016' and '2016a'.
Lines 329-331: Is there a typo here or a missing verb? Is this a conclusion from the comparisons or do we need a reference here?
Line 460 and 560: There is no DOI for the references 'Li and Liu (2018)' and 'Zhang X. (2020)'. Are these confidential reports?

Figures:
Figure 1A: 'Venn diagram of oral microbial genes shared between the different groups'. Why did the authors not keep the term OTU as it is written on line 151?
Figure 1 D-F: The ANOSIM figures are unclear as they sometimes display different colors for the same altitude group. Why didn't the authors keep the same color for each group between the figures?
Figure 2 B-D: Same question here about the same color code for the three different groups.
Figure 3 C: there is a typo in the title « 95% confidence intervals”
Figure S2: the resolution of the figure is unfortunately not good enough to allow us to read the different genera when zooming in on the Network analysis.

---

## Round 0.2 · Minor Revisions

After the first round of revision, there are still some minor concerns that need to be addressed. I strongly encourage you to consider the reviewers' comments, specifically the comments from reviewer 2, to improve your work and manuscript and to resubmit the manuscript.

Reviewer 1 ·

Basic reporting

no comment

Experimental design

no comment

Validity of the findings

no comment

Additional comments

The authors addressed all my concerns; the manuscript is ready for publication.

Reviewer 2 ·

Basic reporting

The authors have addressed some of the comments/critiques however I still have concerns that were not addressed or clarified in the revision process.

1. The introduction is improved however the link between changes in the oral microbiota and some of the more severe diseases (e.g., tumor and cancer development) seems correlative at best. The mechanism by which the oral microbiota can influence these conditions is still unclear. Can the authors provide more direct links to elucidate this connection? Alternatively, I suggest the authors reword these connections to clarify that they are relational rather than causative.

2. Throughout the manuscript the authors refer to different time points as groups (i.e., pre-altitude and altitude groups). This should be revised to avoid confusion as these were not independent groups, rather the same group sampled at different time points (or under different conditions).

Experimental design

1. The authors removed the mention of the non-parametric tests, but in their rebuttal mentioned that a non-parametric test was used to analyze some variables. When and why were these tests used should be clarified in the methods section.

2. The authors provide a justification for their sample size in their rebuttal. What was this estimate based on? Estimated effect sizes? Given that this is a preliminary study it is understandable to not have a power analysis, it should just be clearly stated in the methods.

Validity of the findings

1. In the revised manuscript the authors included a vague mention of the subject’s physical activity. What exactly did the subject do during their time spent at altitude? If they were “walking” much more information is needed to rule out the potential confounding effect of exercise. What was the intensity/duration? Did all subjects perform a comparable amount of activity? This should also be included in the discussion section as a potential limitation.

Additional comments

The manuscript should be thoroughly reviewed for grammar and spelling. Please see below for a few additional comments.

Line 46: “metres” should be meters
Line 47: there is a period missing after “hours”
Line 73: remove the phrase “more and more”. In addition, this sentence needs a reference (or several).
Line 78: This seems like a good point to insert a paragraph break. I am unsure how the remainder of the paragraph relates to the first. In fact, is the second half of the paragraph actually needed? Could it be shortened and still convey the same message?
Line 277: I suggest rephrasing to “…and may contribute to the development of oral diseases”
Line 309: This seems like a good time to insert a paragraph break.
Line 380: The first two sentences in this paragraph need to be revised. Should these sentences be combined?

·

Basic reporting

I would like to thank the authors for taking into consideration my comments and suggestions. The new references and comments have improved the quality of the manuscript. The new figures are also more understandable now.

Experimental design

No comments

Validity of the findings

No comments

Additional comments

Line 167: (Caporaso et al., 2010) is cited but is deleted from the list of references. Is (Sevinsky et al.,2010) the right reference instead? This last one is on the list but I don't find it in the manuscript.

Line 389: the correct term is "saturation diving"

---

## Round 0.3 · Minor Revisions

Dear Tang et al.

The revision you performed needed to be completed. Please make sure all reviewers' requests are appropriately addressed. If you disagree with the suggestion, please, make sure you explain why in your rebuttal letter.

Reviewer 2 ·

Basic reporting

1. The statistics section needs further revision. I think that most of the confusion is related stems from issues with English language. Please revise so that it is clear what statistical tests were performed and when/why they were used.

2. The information provided regarding the physical activity of the subjects is insufficient. The authors states, "The intensity and duration of their physical activity were similar." How was this measured or confirmed? This should be clearly stated in the manuscript. If you are unsure of the actual intensity and duration (e.g., if these data were not captured) then you should state this clearly.

Experimental design

no comment

Validity of the findings

no comment

·

Basic reporting

No comment

Experimental design

No comment

Validity of the findings

No comment

Additional comments

No comment

---

## Round 0.4 · accepted · Accept

The authors have addressed all the reviewers' comments, and the manuscript is ready for publication.